# Stopping Criterion during Rendering of Computer-Generated Images Based on SVD-Entropy

**DOI:** 10.3390/e23010075

**Published:** 2021-01-06

**Authors:** Jérôme Buisine, André Bigand, Rémi Synave, Samuel Delepoulle, Christophe Renaud

**Affiliations:** University of Littoral Côte d’Opale (ULCO), LISIC, BP 719, 62228 Calais CEDEX, France; bigand@univ-littoral.fr (A.B.); remi.synave@univ-littoral.fr (R.S.); samuel.delepoulle@univ-littoral.fr (S.D.); christophe.renaud@univ-littoral.fr (C.R.)

**Keywords:** Singular Value Decomposition (SVD), SVD-Entropy, recurrent neural network, perceptual noise characterization, global illumination, Human Visual System (HVS), stopping criterion, no-reference image quality assessment

## Abstract

The estimation of image quality and noise perception still remains an important issue in various image processing applications. It has also become a hot topic in the field of photo-realistic computer graphics where noise is inherent in the calculation process. Unlike natural-scene images, however, a reference image is not available for computer-generated images. Thus, classic methods to assess noise quantity and stopping criterion during the rendering process are not usable. This is particularly important in the case of global illumination methods based on stochastic techniques: They provide photo-realistic images which are, however, corrupted by stochastic noise. This noise can be reduced by increasing the number of paths, as proved by Monte Carlo theory, but the problem of finding the right number of paths that are required in order to ensure that human observers cannot perceive any noise is still open. Until now, the features taking part in the human evaluation of image quality and the remaining perceived noise are not precisely known. Synthetic image generation tends to be very expensive and the produced datasets are high-dimensional datasets. In that case, finding a stopping criterion using a learning framework is a challenging task. In this paper, a new method for characterizing computational noise for computer generated images is presented. The noise is represented by the entropy of the singular value decomposition of each block composing an image. These Singular Value Decomposition (SVD)-entropy values are then used as input to a recurrent neural network architecture model in order to extract image noise and in predicting a visual convergence threshold of different parts of any image. Thus a new no-reference image quality assessment is proposed using the relation between SVD-Entropy and perceptual quality, based on a sequence of distorted images. Experiments show that the proposed method, compared with experimental psycho-visual scores, demonstrates a good consistency between these scores and stopping criterion measures that we obtain.

## 1. Introduction

Realistic image computation mimics the natural process of acquiring pictures by simulating the physical interactions of light between all the objects, lights and cameras lying within a modelled 3D scene. This process is known as global illumination and was formalised by Kajiya [1] with the rendering Equation (Equation 1):(1)$$L_{o}\left(x,\omega_{o}\right)=L_{e}\left(x,\omega_{o}\right)+\int_{\Omega }^{}L_{i}\left(x,\omega_{i}\right)\cdot f_{r}\left(x,\omega_{i}\to \omega_{o}\right)\cdot \cos\theta_{i}d\omega_{i}$$
where (see Figure 1 for the annotations):$L_{o}\left(x,\omega_{o}\right)$ is the luminance traveling from point *x* in direction $\omega_{o}$;$L_{e}\left(x,\omega_{o}\right)$ is point *x* emitted luminance (it is null if point x does not lie on a ligth source surface);the integral represents the set of luminances $L_{i}$ incident in *x* from the hemisphere of the directions Ω and reflected in the direction $\omega_{o}$ (two of these directions are shown in Figure 1 as $\omega_{i1}$ and $\omega_{i2}$). The reflected luminances are weighted by the materials reflecting properties (bidirectionnal reflectance function $f_{r}\left(x,\omega_{i}\to \omega_{o}\right)$) and the cosinus of the incident angle.

This equation cannot be analytically solved and Monte Carlo approaches are generally used to estimate the value of the pixels of the final image.

Sampling is performed through the build of random light paths between the camera and the light sources lying in the 3D scene: a ray is sent from the camera location through a pixel and is randomly reflected by the surface of the first object it encounters. The process is recursively performed until a light is reached or a Russian roulette process decides to stop the path. Numerous light paths are build per pixel and according to the law of large number and the Monte Carlo approach, the mean of the samples for each pixel converges to the solution following a $\frac{1}{\sqrt{n}}$ rate where *n* is the number of samples [2]. Unfortunately convergence requires often several hours (even days) before a visually usable image is available, due to both the complexity of paths computation and the high number of samples that are required. Furthermore information about the number of paths that are really required for the image to be visually converged is unknown. Stopping computation too soon provides images with visual noise and computing too much samples can lead to a loss of time and higher production costs. Figure 2 gives such an example of perceived perceptual noise with a decreasing level of visual noise from left to right images. Automatically deciding when visual convergence is reached for each part of an image is thus a real challenge. In this paper, a “No-reference” specific technique which takes into account the amount of perceptual noise to establish image quality and to provide a computation stopping criterion is introduced.

We therefore propose to acquire human subjective thresholds from numerous computer-generated images with different noise levels. Specific characteristics based on the entropy of the Singular Value Decomposition (SVD) decomposition are extracted from all noisy/noiseless images. These characteristics are then used in a learning process that provides us with a model capable of deciding whether an image still contains visible noise. This approach allows expensive rendering engines to automatically decide whether the calculation can be stopped or should be continued.

Based on these ideas, the work presented in this paper will be organised as follows: first, different approaches to image quality assessments that could be applied to our problem will be recalled. Then, a database of synthetic images and subjective human thresholds collection will be detailed. Section 4 will present the SVD-Entropy [3] that is an interesting tool for noise quantification in photo-realistic synthesized images. Then the proposed methodology for noise detection using SVD-entropy and a Recurrent Neural Network (RNN) Deep Learning technique will be introduced and the obtained results will be detailed before proceeding to a conclusion and perspectives of the paper.

## 2. Previous Works

As presented in Introduction, realistic image computation is based on the Monte Carlo (MC) techniques. Thus the perceptual noise that affects computer-generated images (CGIs in the sequel) is not easy to model. Another challenge concerning CGIs is that a reference image is not available at the beginning of the process, and it is difficult to forecast a stopping criterion. We now present some existing techniques to deal with some aspects of these complex issues.

### 2.1. Improving the Convergence of Rendering

To date, the rendering community has focused mainly on accelerating the convergence of Monte Carlo sampling and, more recently, on denoising approaches.

Since the first path-tracing rendering algorithm proposed by Kajyia [1] in 1986, several new integrators have been designed. Their purpose is to carry out a ray-tracking strategy in order to strategically improve the contributions within the scene for each pixel and consequently reduce more quickly the visual noise. Among them, the bidirectional path-tracing [4] builds paths from both the camera and the light sources and attempt to connect the different path nodes, increasing the quality of the light energy collection. Metropolis Light Transport [5] constructs paths from the eye to a light source using bidirectional path tracing and then applies slight modifications to the paths in order to explore the neighbourhood of this one. Some careful statistical calculation (the Metropolis algorithm) is used to compute the appropriate distribution of brightness over the image.

Sampling strategies (also called adaptive sampling or adaptive rendering), where the remaining samples are distributed over the pixel grid more intelligently than randomly have been studied. In [6], Non-local means filtering algorithm is used for denoising the image and predict the sampling map (next samples allocation for each pixel) using difference between the current original image and the denoised one. The Stein’s Unbiased Risk Estimator (SURE) which offers a mean for estimating the accuracy of a given estimator was also used for adaptive sampling and reconstruction in rendering and seems to give interesting results [7]. In [8], a new sampling and reconstruction method adapted to the image plane, based on local regression theory, has been proposed and gives better results than previous methods.

Path-guiding techniques [9] have also been proposed; they aim to target in a more intelligent way the areas that seem the most interesting for lighting or that requires more calculations. In this way, path and its rebounds are orientated towards areas with a high contribution into the scene. This can be done through learning algorithms by reinforcement thanks to extracted statistics or other scene information.

In the context of our study we do not target explicitly a faster rendering technique, but rather a way to stop the computation in any part of an image when it is considered as noiseless and this independently of the chosen unbiased integrator. Of course a direct consequence of such a goal will be to reduce the computation time compared to a “brute force” approach, but we focus mainly on the understanding of human perception of noise and, therefore, on image quality measurements.

### 2.2. Study of Noise in Different Kind of Image

#### 2.2.1. Image Quality Measures

For images of natural scenes, there are a large number of methods (called Image Quality Assessment-IQA) that allow the image quality to be calculated from a model that has learned from subjective data.

Therefore, objective IQA methods have been widely studied [10,11,12,13]. According to the availability of a reference image, objective IQA methods can be classified as full-reference (FR), reduced-reference (RR), and no-reference (NR) methods. In a FR method, a distortion is applied on an original image providing a distorted one and the assessment result is obtained through the comparison of the two images. With the advances of recent studies, the accuracy of this kind of method is getting better, despite its disadvantage of requiring a complete reference image, which is often not available in practical applications. Known metrics such as Structural SIMilarity metric (SSIM), Multiscale SSIM (MS-SSIM) [14] and Peak Signal to Noise Ratio (PNSR) are well known for this approach, both of them requiring the reference image. RR methods, which are also known as partial/reduced reference methods, do not make a complete comparison between the distorted image and the original one, but only compare extracted or available features. NR methods, also called blind image quality assessment (BIQA) methods, don’t require any image as reference. The evaluation score is therefore obtained using only the distorted image.

Unfortunately as mentioned in [15], it appears that well-known image quality metrics such as SSIM, MS-SSIM, PNSR seemed to show weaknesses in relation to the MC noise present during image generation. In fact, they are often tuned for compression/transmission artifacts and have not been evaluated in the context of synthetic computer generated images.

Finally works focused on image quality measurement require image databases where the distortions applied to the natural images are known such as LIVE [16] and TID [17]. Such databases are also available for computer generated images [15,18] but the provided information does not allow us to use it to measure the MC noise of generated images during rendering. We will therefore describe the content of a new image database specifically built for the analysis of the perception of Monte-Carlo noise in computer-generated images.

#### 2.2.2. Denoising Techniques

One of the interests of the IQA is to make it possible to measure the effects of noise reduction methods that have been widely studied in the context of natural images. The particularity of these methods is that they are designed to reduce additive noise flawing the reference image (Salt-and-pepper, Gaussian, Shot noise, etc.) or artifact linked to a loss of information (JPEG compression for example).

Among filter based methods, there are two main approaches, (i) methods oriented on the spatial domain (filters work directly on pixels of image) [19,20] and (ii) methods based on transform domain such as wavelet transform domain [21,22,23,24]. More recent works are focused on deep learning approaches such as Convolutional Neural Networks (CNN) for denoising natural images [25,26].

Some existing techniques in Video quality assessment could also be considered to manage our objectives. Indeed Video IQAs are rather far of classic IQAs [27], since they use spatio-temporal metrics, discussed in VQEG, and they are mainly devoted to compressed videos (MPEGs). Some recent works, more oriented to CGIs [28], and using a specific dataset (MOAVI) have been presented. Nevertheless these works are devoted to specific artifacts, and the MOAVI project seems to be difficult to apply for the detection of Monte Carlo noise at the moment, thus these techniques have not been used. Moreover the pipeline presented in the paper (based on a reservoir of following images) is quite different from a video sequence.

Post processing methods are also proposed in the Computer Graphics domain, in order to reduce noise in the image at the output of the rendering engine. The more recent methods for such denoising task exploit mainly Deep Learning [29,30,31,32,33]: through an Auto-Encoder model, both the information from the current image and that extracted from the scene allow the model to learn how to restore the image to a so-called reference image (computed with a very large number of samples).

Our work is not focused on image noise reduction but on Monte Carlo noise quantification and detection, thus denoising techniques have not been studied.

#### 2.2.3. SVD Based Noise Quantification

Concerning noise quantification, some methods use the Singular Value Decomposition (SVD) which is a very powerful matrix factorization method. SVD is thus used for features extraction in order to obtain a noise estimation of the image and process to its reconstruction. The approach introduced in [34] is based on the fact that the smallest values of the singular value vector would be representative of the noise, which facilitates its detection and its filtering. It is then extended to a blockwise slicing of the image, called Block based SVD (BSVD), in order to facilitate the calculations of the SVD decomposition [35]. The idea is to (i) apply a Gaussian filter to the noisy image, (ii) subtract the filtered image from the noisy image to obtain an image containing the contours of the objects and the noise, (iii) apply BSVD to the latter image to remove the noise, and finally (iv) add the filtered image and the noise-free contour image to obtain the original noise-free image. According to [36], the SVD decomposition seems to decompose the structure of the image (first part of the components of the SV vector) from the remaining attributes (independent of the structure and coming from the other components of the SV vector). The low components of the singular value vector that are the most sensitive to noise are then used to estimate the quality of an image or estimate noise level [37,38,39].

### 2.3. Stopping Criterion

In the case of computer-generated image field, the nature of the noise is an inherent part of the generated image, and also linked to the scene itself. Using known natural image IQA is thus not possible and according to our knowledge, databases built from computer generated photo-realistic images for MC noise study do not exist. In the context of NR prediction, some methods have been proposed based on perceptual subjective human data. For each of the methods, the model tries to predict whether or not the image is still noisy for human.

In [40] images of size 512 × 512 have been divided into 16 parts of size 128 × 128. For each part of the image, a human threshold (number of samples per pixel) was obtained from experiments. Then 26 attributes are extracted from the L-channel of the Lab color space of the processed images, with different number of samples. These attributes come from well-known filters applied to the image: linear and Gaussian filters, median and Wiener filters with a window size of 3×3 and 5×5. A wavelet decomposition is also applied to reconstruct the image without the low spatial frequencies of L. Following this way, 13 filtered images are obtained through the applied filters and the use of wavelets. For each of these images, the mean error and the standard-deviation of error are extracted. The error is obtained from the difference between the features of the test sub-image and a quick ray traced sub-image of the scene (assumed to be the reference image). These values form the 26 input noise attributes of a SVM model in order to classify image as noisy or not. Authors also proposed an approach [41] where the previously proposed filters are applied one after the other on the input image. The frequency bases are then extracted from this filtered image and a noise-free image is obtained. The difference between the original image and the noise-free image, so-called “noise mask” is used directly as the input to the model, i.e., a vector of size 128×128. Another approach proposed in [42], consists in adding samples to the input image using path tracing algorithm in order to obtain an image considered as a reference. For each of the two images (input image and approximate reference) a blurred image is computed using a 3×3 Gaussian convolution with a convolution coefficient ∈[0.3,1.5]. Then, two noise masks are obtained by computing the difference between the original image and the blurred one for each of the images. The two noise masks are then used as input to an SVM model to predict the label of the input image. These methods both use a kind of reference image and are not exactly NR-IQA. Thus a generic NR method, based on the intrinsic features of the image, is presented in the sequel. Before that, the next part of the document is devoted to the method used to collect human subjective data, which will enable the noise present in the images to be quantified.

## 3. Subjective Dataset Collection

In the computer-generated image field, the nature of the noise is specific to the Monte Carlo approach, but also to the scene itself. It is thus very different of well-known noise models and using available natural image databases is not possible. Then data that were used in [41,42] are still limited both in terms of image size and scene complexity.

One of the objectives of this work was thus to build a huge dataset of photo realistic synthetic images with different levels of noise, and to provide subjective human thresholds of noise perception for each of them. Several points of view of commonly used 3D scenes [43] were calculated with several levels of samples. The rendering engine used was PBRT (for Physically Based Rendering) version 3 [44] and only one integrator (Path Tracing) has been used among the various available, both because of its wide use and to avoid the biases that could arise when using different integrators.

Our image data base therefore consists of 40 different images with a per pixel sampling ranging from 20 samples to 10,000 with a 20-samples pitch. An overview of the reference images of these 40 points of view is presented in Appendix A. These are the images calculated with the highest number of samples per pixel and are considered to be noise-free. Several types of scenes have been chosen, representing various geometries and for which different types of materials are present to provide various lighting effects. Each image is of size 800×800, this size having been chosen to allow two images to be displayed side by side on standard screens. As with the methods proposed in the literature, the images are cut into 16 non overlapping blocks, each measuring 200×200 pixels (see Figure 3). The human experimental data have been gathered for each block of each image, in order to get human thresholds of visible noise in each of them.

In order to obtain these subjective human thresholds, an application has been developed that initially shows the most noisy image on the left, and the so-called reference image on the right (the 10,000 samples image). It can be noted here that the procedure applies in a so-called “Just noticeable difference” (JND) framework, where the user is assumed to perceive or not perceive a difference between two images. During the experiment, the user is asked to point the locations where left and right images are different. At each click-to-point operation, the corresponding block is identified and is replaced in the left image by a block with more samples (see Figure 4). Block sampling is increased by steps of 20 samples. When the user considers that both left and right images match, the different sampling levels are recorded and are assumed to be its visual thresholds of noise perception. The mean of the scores obtained for all the users also called Mean Opinion Score (MOS) is then considered as the human threshold and will be used in our studies.

An example of block cutting and subjective thresholds obtained per block is shown in Figure 5. The human reference image is reconstructed from the human subjective thresholds obtained for each block.

From those subjective human thresholds, it is finally possible to provide a label (noisy or noiseless) to each sub-image of any 800×800 image: sub-images in a block whose sampling level is less than the human threshold are considered as noisy, and noise-free in the opposite case. Figure 6 shows how the labels are assigned for each block in the image.

Once the collection of subjective datasets has been built up, a new NR-IQA dedicated to computer-generated images can be developed. The next section aims to introduce the SVD-entropy measure used in our approach. The presented technique makes it possible to link SVD-entropy to perceptual quality. It should be noted here that the notions of image quality and stopping criteria can be quite the same. Indeed, in this work, the image quality is representative of the number of necessary samples from which humans no longer perceive the noise.

## 4. SVD-Entropy

Entropy is a general framework to extract information in a noisy environment. Particularly image entropy is a statistical feature that reflects the average information in an image and can be used as a measure for image quality. In the case of SVD of an image L, an entropy measure can be defined, namely SVD-entropy that makes it possible to quantify the perceptual noise inherent to computer-generated images as shown now.

### 4.1. Overview of Singular Value Decomposition (SVD)

The proposed method is based on SVD which is a very powerful matrix factorization method. In the theory of SVD [45,46], any M×N image *L*, with rank(L)=O, where O=min(M,N) (*L* is a rectangular real-valued matrix built from the L-channel of the image in the $L_{ab}$ color space), can be uniquely decomposed as:(2)$$L=U\sum V^{T}=\sum_{p=1}^{O}\sigma_{p}\vec{u_{p}}\vec{v_{p}}$$
where *U* and *V* are respectively M×M and N×N orthogonal matrices with column vectors $\vec{u_{p}}$ (called the left singular vectors) and $\vec{v_{p}}$ (called the right singular vectors) and ∑ is a diagonal matrix containing the square roots of eigenvalues of $LL^{T}$ ($\sum =diag(\sigma_{1},\sigma_{2},...,\sigma_{O})$). The diagonal elements of ∑ can be arranged in a descending order and are called the singular values of *L*. SVD is a reliable orthogonal matrix decomposition method. In the sequel, the matrix *L* will correspond to an image defined using gray levels (luminance component of any color image $L_{ab}$). SVD is built from two orthogonal dominant and sub-dominant subspaces. Each singular value (SV) relies to the luminance of an image while the corresponding pair of singular vectors (SCs) relies to the geometry of the image. This attractive property is used in noise filtering, watermarking, image compression, …, [34].

### 4.2. Noise Features Based on SVD

The proposed model is based on well-known relations between image noise and SVD. Ref. [47] have shown that the largest object components in an image generally correspond to images associated with the highest singular values (SVs in the sequel) $\sigma_{i}$, while image noise corresponds to images associated with the lowest SVs. Thus, through the psycho-visual properties, the visual quality of the image is considered not to be significantly reduced by ignoring the lower singular values [48].

Image noise appears as an increasing spatial activity in spatial domain of the image that is linked with the lower values in SVD domain [34]. Therefore, when noise is added, singular values are non-uniformly increased: the medium values are increased by largest amount, although lowest singular values have the largest relative change. This property is illustrated Figure 7, where the (normalized) norm of the 800 SVs of the Living room 3 image in Figure 2 are computed for different steps of quality convergence.

As mentioned in [39], the first quarter of the components of the SV vector seem to be related to the structure of the image and the rest of the components to the noise present in the image. Figure 8, illustrates this “property” after reconstruction of the grey level images of the L channel of the $L_{ab}$ decomposition. The reconstructed image is quite close to the original image when only the first 25% of the elements are used.

### 4.3. SVD-Entropy as Noise Feature

Considering the singular values of the image *L*, computed as a matrix (*O* is the rank of the matrix *L*, $\sum =diag(\sigma_{1},\sigma_{2},\ldots ,\sigma_{i}\ldots ,\sigma_{O})$) the relative significance of the ith SV in terms of the fraction of the overall expression that they capture is expressed as:(3)$${\bar{\sigma }}_{i}=\sigma_{i}^{2}/\sum_{p=1}^{O}\sigma_{p}^{2}$$

SVD-entropy ($H_{SVD}$) is defined in Equation (Equation 4) using relative components significance (Equation (Equation 3)). It measures the complexity of the data from the distribution of the overall expression between the different SVs, where $H_{SVD}=0$ corresponds to an ordered and redundant dataset (all expression is captured by a single SV), and $H_{SVD}=1$ corresponds to a disordered and random dataset where all SVs are equally expressed.
(4)$$H_{SVD}=-\frac{1}{\log(O)}\sum_{i=1}^{O}{\bar{\sigma }}_{i}\log\left({\bar{\sigma }}_{i}\right)$$

In our problem, $H_{SVD}$ may be an indicator of how image is still composed of noise. Since during the rendering of an image the number of samples increases, the SVD-entropy should make it possible to indicate how much noise still affects the image. It could also be an indicator of the stability of the image during its generation (convergence). So SVD-entropy [46] makes it possible to compute the most effective noise features and thus to determine the most efficient scheme for computer image generation. Consider the SVs of the previous computer-generated image (Living room 3). Figure 9a shows the evolution of $H_{SVD}$ for the Living room 3 image at different sampling levels.

Looking at the local area information, i.e., the different blocks used in the dataset, the amount of noise picked up by the SVD-Entropy of noise is very different from one block to another. Figure 9c shows that the value of $H_{SVD}$ and its progression differ from one block to another. When normalising each block (see Figure 9d), human thresholds seem to better fit with $H_{SVD}$ convergence.

Based on the elements found from the SVD-Entropy, the following section will present how it has been used within an Recurrent Neural Network (RNN) model to consider whether or not a block in the image requires additional computation.

## 5. The Proposed Method

We propose a two-stage method based on SVD-entropy and Recurrent Neural Networks, RNNs (the general pipeline is detailed in the following).

### 5.1. Recurrent Neural Networks

RNNs are a class of artificial neural networks where the connections between nodes form a directed graph along a time sequence. This allows it to mimic a dynamic temporal behaviour. Derived from feedforward neural networks, RNNs can use their internal state (memory) to process variable length sequences of inputs. They are now widely used since several types of tasks can be applied to them once the architecture of the network is well defined [49,50,51]. For example, they are used in natural language processing for translation, sentence auto-completion (prediction of the next word), but also for text classification [52] or text generation [53]. Depending on the nature of the information, these tasks can be of a different nature and not only focused on word processing. Videos in particular can be considered to be a sequence of images from which it is possible to extract information and classify some elements [54,55].

When rendering an image through a Monte Carlo process, it is possible to consider each step of the image enhancements (adding *n* samples) as a sequence of images reviewer (or image reservoir): the first image of the sequence is the most noisy, obtained during the first steps of the computation; the last image corresponds to the noise-free image (from a perceptual point of view), obtained at the end of the computation. Thus, we propose to keep the last *k* computed images, each one with an increasing number of samples, to extract information related to the noise in these images, then to provide this information to a network in order to know if the last image of the sequence is still noisy. The well known Long Short Term Memory (LSTM) cell [56] is used for this task with the aim of making the best possible use of previous information from the images in the sequence before estimating the label. The model would therefore be learnt in a so-called supervised learning environment based on the data collected through the experiments carried out. Figure 10 describes the general architecture of the RNN network where *k* images with a sampling step of n=20 between each of them are given as input.

### 5.2. Noise Features Extraction

Feeding such a neural network with only the value $H_{SVD}$ for each block of sequence of size *k* did not seem relevant from the point of view of the first experimental results. To allow the model to correctly learn the noise in a block, a split was performed again, providing *m* sub-blocks for which the value $H_{SVD}$ was extracted. If the block image of size 200×200 is cut into 25 sub-blocks of size 40×40, then the model will have at its disposal a vector of 25 features for each block, where each one is a $H_{SVD}$ value extracted from the sub-block $m_{i}$ with i∈[0,25]. There are therefore *m* features extracted for each block image in the *k* size sequence. The underlying intuition is that the model can interpret the evolution of noise in each sub-block through the sequence of characteristics. This would amount to proposing a bottom-up model, where local information is used to predict global information. In our case, the question is whether a part of the image can be considered as noisy or not. In summary, the features that will be used as RNN inputs will be formed from the entropy values associated with each of the SVD vectors from the *m* sub-blocks of a 200 × 200 image block (see red frame in Figure 11).

## 6. Experimental Results

In this section, we describe the studies we conducted to evaluate the proposed approach. We begin by identifying the various parameters whose values could have an impact on the results. We then describe the testing methodology and the effectiveness measures that will be used to evaluate the approach. Finally, we exhaustively detail the results we obtained, both on the learning image database and on images unknown from the best model obtained.

### 6.1. The Method Parameters

In relation to the input features of the model, several parameters will be studied:The size of the sequence of RNN with k∈[3,4,…,10];The number *m* of sub-blocks per image block, with m∈[4,25,100,400] and hence the extracted vector size of sub-block image. Sub-blocks size are respectively of size 100×100, 40×40, 20×20 and 10×10;The sampling step of images to pass to the RNN sequence input with n∈[20,40,80].

Two additional parameters were studied, related to the choice of the components of the singular values vectors to be used and their normalization.

As mentioned in [39], the so-called 3/4 low value components of SV appear to be linked to noise. The question that then arises is whether to use the entire SV vector, the first quarter or to restrict itself to the last 3/4 of its values. In these last two cases, the corresponding reduced entropy measures are respectively called $H_{SVD}^{1}$ and $H_{SVD}^{2}$ and are defined by Equations (Equation 4) and (Equation 5). The value extracted from a sub-block is then defined by a new parameter $F\in [H_{SVD},H_{SVD}^{1},H_{SVD}^{2}]$.
(5)$$H_{SVD}^{1}=-\frac{1}{\log(\frac{O}{4})}\sum_{i=0}^{O/4}{\bar{\sigma }}_{i}\log_{2}{\bar{\sigma }}_{i}$$
(6)$$H_{SVD}^{2}=-\frac{1}{\log(O-\frac{O}{4})}\sum_{i=O/4}^{O}{\bar{\sigma }}_{i}\log_{2}{\bar{\sigma }}_{i}$$

The normalization of input features was also studied. Two types of normalization were tested: one where the features vector extracted from a sub-block of the image is normalized before the vector is inserted into the input sequence of the RNN; the other where each feature is normalized within the sequence itself, in order to study the evolution of the feature for the sub-block (normalization process is presented in blue frame in Figure 11). These ways of normalization are named here respectively bnorm (block-normalization) and snorm (sequence-normalization). Note that the whole sequence data are required to compute snorm. The combination of the two kind of data normalization was also studied, bnorm=1 for each image features and snorm=1 when passing sequence data into the RNN model. However normalising by sequence could make it possible to find a slope for each sub-block, as observed in Figure 9d. The model would therefore have information on whether or not *F* converges on the *k* image sequence for each sub-block. The pipeline of the proposed method is detailed in Figure 11.

### 6.2. Model Comparisons

The 40 images available in the image database are divided into 16 non-overlapping blocks, of which 12 blocks are randomly selected for the learning and validation set and the remaining 4 for the test set. The idea is to allow the model to learn as much as possible from the different scene structures. The selected areas are saved and reused for each new learning of a model, whether they are of the RNN type or those from the literature. The aim is to compare the performance of each model as fairly as possible.

To compare the performance of a model, the well known metric of “Area Under the Receiver Operating Curve (AUC ROC)” [57] is used. ROC curve plots true positive rate vs. false positive rate at different classification thresholds. Lowering the classification threshold classifies more items as positive, thus increasing both false positives and true positives. AUC provides an aggregate measure of performance across all possible classification thresholds. Hence, AUC score is a performance measurement for classification problem. It indicates the extent to which the model is able to distinguish between classes.

### 6.3. Results

All parameters combinations have been analysed in order to find the best parameters expected on RNN with LSTM cells. Our model is composed of 3 LSTM layers with the following respective layer sizes: 512, 128, 32 and a drop-out rate of 40% on each one in order to avoid overfitting. Each LSTM layer has a Sigmoid activation function for cell state and hidden state. A Hard Sigmoid activation function is set to activate the input/forget/output gate. A dense layer of size 1 is then used with the Sigmoid activation function as output in order to predict the expected label. Then the binary cross-entropy loss is employed to propagate the network error during training. Data balancing has also been added to prevent one class from another to be dominated according to the distribution of data in the learning base. The loss function is weighted when propagating error, which leads to a higher weight for a sample with a less frequent label.

As it is impossible in terms of memory cost to give all the learning data, it is preferable to use batch. Batch size defines the number of samples that will be propagated through the network in order to update the weights of the whole network using loss function for a gradient based regression process. When learning RNN, this parameter has been also studied, batch size was fixed with $b_{s}\in \left[64,128\right]$, to see the impact of batch in our case on loss function when propagating the error.

#### 6.3.1. Training and Testing Results

For each of the tested combinations, the same datasets are processed and the output model is trained over 30 epochs. Table 1 displays the 20 best model performances obtained. For the parameters snorm and bnorm, the value 1 means that the corresponding normalization has been applied, 0 otherwise.

When observing the results, several important things become apparent. First of all, using the entire SV vector before calculating entropy seems to bring better results. Indeed, none of the best models include the use of the reduced entropy measures such as $H_{SVD}^{1}$ and $H_{SVD}^{2}$. The interpretation that can be made behind this is that the model needs as much information as possible, even if the proportion of noise on the singular values of highest modulus could appear to be low. Figure 12 shows an overview of reduced $H_{SVD}^{1}$ and $H_{SVD}^{2}$ over the same selected zones that were used in Figure 9.

$H_{SVD}^{1}$ is very closed from the use of whole SV components (see Figure 9d). When using $H_{SVD}^{2}$ on image blocks mainly receiving indirect lighting (blocks 1, 11 and 13), the entropy remains high due to the fact that the lower weight components are very sensitive to noise. Using all components with $H_{SVD}$ seems to give both noise and structure information to the model, allowing it to learn better. The scene structure information that is mainly available through the lower order singular values can allow the model to differentiate between scenes and generically allows for noisy and noiseless differentiation. It should be noted that noise is also present in this lower part of the SV vector even if its magnitude is much smaller than the corresponding singular values. This could increase too the available information during the learning process. The number of sub-blocks m=100 seems a good compromise in terms of information: the exploration of local information appears to give more details on the image for the network. On the contrary, too much information, as with m=400, appears to reduce the ability of the model to interpret the data properly.

Data normalization also seems to work much better when normalizing the sequence (snorm) feature by feature rather than using bnorm or a combination of both (bnorm = 1 and snorm = 1). The intuition behind this comes from the fact that each sub-block of the input image is processed independently, which gives the model a better understanding of the evolution of noise in the image sequence.

More generally, the top 60 results obtained on the different combinations of parameters tested provided scores between 79% and 82.5% on the AUC test, using mainly $F=H_{SVD}$, snorm=1 and m=100. The sole exceptions was the the 40th best result, for which AUC score was 80.2% with m=40. bnorm=1 seems to give the worst results and shows that the model does not interpret the data for this kind of normalization. The best results when using $F=H_{SVD}^{1}$ and $F=H_{SVD}^{2}$ are respectively 79.12% and 65.83% which highlights the importance of calculating the entropy on all the components of the SV vector on each sub-block.

Another very important point here is that the model does not seem to overfit, which is very important for future predictions about unlearned blocks of images. This generalisation also makes it possible to show the consistency of the data on the contribution of noise detection.

The first line in Table 1 indicates the selected RNN model used to simulate the prediction of human subjective thresholds on certain points of view. Looking at the sequence size parameter *k*, and starting from the others selected parameters m=100, $F=H_{SVD}$, $b_{s}=128$, n=40 as samples step and snorm as normalization process. Among the best models obtained, we study the evolution of *k* and its impact on the performance of the model It would seem that the model increases more significantly with a ROC AUC score evolving from 80.5% to 82% when k∈[3,5]. Afterwards, it evolves more slowly until a peak of convergence towards k=8 (with 82.5%) and then decreases very slightly down to 82.22% for k=10 which was the last sequence size we tested. Results thus appear to be low sensitive to the value of parameter *k* in the interval [5, 10].

#### 6.3.2. Comparisons

The methods that have been previously proposed for issuing a stopping criterion when rendering are all based on a pre-processed image that is used as an approximation of the converged image. The main disadvantage of this approach is that the calculated image does not necessarily include certain complex effects that are usually important in lighting simulation and noise distribution. The approach we propose is a no-reference one, in the sense that only the images being computed are used. All the light effects are taken into account by the simulation process and a comparison with the approaches with reference would probably not make sense, due to their intrinsic limitations.

However we indirectly compared our approach with that of Constantin [40] in order to determine whether the features extracted from the SVD-Entropy were consistent for characterizing the noise. The idea is to use the 26 attributes they proposed in place of our SVD-entropy features. For each of the *k* images, the 26 attributes are extracted and transmitted to the RNN model. Because of the temporal aspect taken into account by this type of network, the differences between the *k* sets of 26 attributes will be implicitly managed by the RNN. This approach seems fairly close to Constantin’s way of operating, without requiring the use of an approximate reference image, which is more or less distant from the image that will be calculated by the rendering process (the 26 attributes are extracted from the difference between the approximate reference image and the noisy image being calculated).

Table 2 shows the results of combinations of the same model parameters using the 26 attributes proposed in [30] as input. However, the parameters *m* and *F* are not dealt with here, as they relate to the SVD-Entropy approach. The top 20 models vary with an AUC ROC score from 79.35% to 81.16%, which is slightly lower than the best model with SVD-Entropy based attributes as model input (82.55%). Note here, that the basic normalization (bnorm=1) of the image features themselves (the 26 attributes vector) combined with the normalization by sequence (snorm=1) seems to give correct results (80.59% AUC ROC score for the best model with these parameters in particular). The bnorm normalization is in fact used for reducing to the same order of magnitude features of different scales. This is the case for the 26 attributes as opposed to the SVD-Entropy attributes, which are already all on the same scale (score between 0 and 1). The $b_{s}$ parameter does not seem to have a significant impact on the model. On the other hand, the *n* parameter, representing the number of samples between each element (image) of the sequence, seems to bring the best results to the model when it is relatively smaller (n=20 and n=40). The results also indicate that a fairly large *k* (such as k>7) also seems to give better results because the information is richer. The second line represents the selected model with the input sequence size chosen k=8 in order to well compare with $H_{SVD}$ RNN based model. Others selected parameters are hence $b_{s}=64$, bnorm=0, snorm=1 and n=20 as samples step. As for the $H_{SVD}$ based models, the *k* parameter was observed. The model with the 26 attributes is quite sensitive on the improvement of its performance with a ROC AUC score evolving from 76% to 80.5% for k∈[3,8], before decreasing very slightly towards 80%, so k=8 also seems to be a performance compromise.

#### 6.3.3. Rendering Simulation

Rendering simulations have been processed in order to compare the two selected models, one with $H_{SVD}$ and the other with 26 attributes, as noise input features for each image of the sequence. The procedure for enabling a simulation is carried out as follows: for each level of noise (number of samples per pixel), the model is called to predict each block label. Samples are processed from 20 to 10,000 per pixel using a step of *n* samples, with n=40 for the $H_{SVD}$ model and n=20 for the other one. Once a block is no longer detected as noisy, then the current samples step is saved as threshold.

The RNN model shown in Figure 10, provides as output a prediction probability between 0 and 1. If the probability is less than 0.5, then the image is considered to be no longer noisy, if not, it is still noisy. By analysing the evolution of these predictions during some simulations, a common behaviour could be identified, since the model appears to have some moments of hesitation between the two labels (noisy/not noisy) near the human threshold on a block (see Figure 13). To overcome this problem and to make thresholds prediction more robust, it was proposed to consider that a block is no longer noisy after 3 successive noiseless predictions.

Table 3 shows the performances of each selected model ($H_{SVD}$ and 26 attributes) on all 40 viewpoints (both learning and testing blocks) for which human thresholds were available. The notion of margin of error is introduced, which is defined as the percentage of prediction errors allowed in relation to the maximum number of samples used to obtain the reference image around the human threshold. A margin of 2% thus means that a result provided by the model that lies within the interval [T−0.01×Max,T+0.01×Max] will be considered a good answer, with *T* the human threshold and Max the number of samples for the reference images (here 10,000 samples). This makes it possible to take into account fluctuations in human vision around the average acquired threshold and to estimate the performance of the model as a function of the width of this margin.

The use of this margin of error allows models to obtain more accurate prediction scores, up to more than 88% for a margin of 5%, for both approaches ($H_{SVD}$ and 26 attributes). Of course, these scores increase as the margin of error increases. Nevertheless, it can be noted that the $H_{SVD}$-based model keeps a more interesting prediction rate than the one using the 26 attributes, with an average difference of 3% between the two models.

In some cases the model provides us with predictions of critical thresholds that are surprising. Figure 14 shows a very rare case, which highlights a hesitation of the model during an important time interval and leads it to provide an erroneous threshold estimate. The block targeted here from the bathroom point of view is still noisy up to 10,000 samples (see Figure 14c) and seems to contain a significant light reflection. Unfortunately, the model does not seem to understand this behavior over an image interval (Figure 14). A probable hypothesis concerning this problem could be that this light effect is not sufficiently present in the learning base for the model to be able to interpret it correctly during the convergence phase of this block.

#### 6.3.4. Image Reconstruction

Based on the previous simulations and the thresholds predicted by the RNN with the input $H_{SVD}$ characteristics, the images are reconstructed and compared to the reference images. 6 of these images, among the 40 for which human thresholds are available, are presented in Section B.1 (most noisy images) and Section B.2 (reference images). Figure 15 shows the comparisons between the image reconstructed by the model, the image obtained from human thresholds and finally the reference image. A zoom is performed on some parts of each image. The SSIM metric [14] is used to calculate the difference between each image and the reference.

All model predictions are relatively close to the images reconstructed using human perceptual thresholds, as evidenced by the SSIM values used for comparison. Most of the time, the model predicts a higher SSIM score, with the model often going a little further than the thresholds. When the SSIM value of the image reconstructed from the model is lower than the SSIM value obtained from the perceptual thresholds, the numerical difference remains small and visually undetectable.

The goal of our approach is obviously to be used on new images that the model has never seen before and for which we have no value for the human visual threshold. We have therefore carried out new tests on such images. We present 4 of them in the Section C.1 (most noisy images obtained for these images) and Section C.2 (reference images computed with 10,000 samples per pixel). As before, we use the SSIM measure to evaluate the quality of the reconstructed images after the model has decided to stop the computation of each of the 16 blocks composing them. The results obtained are presented in Figure 16 and show that the model obtained seems robust enough to provide stopping thresholds that no longer allow the perception of noise.

The Figure 17 gives an overview of the thresholds obtained on each of these four images. Even if no human threshold is available, the number of samples predicted by our model is consistent with the objects and lighting present in each block. Areas of scenes with mainly direct lighting require far fewer samples, while areas with mainly indirect lighting require more samples. A difficult case of indirect lighting is illustrated by the San Miguel scene, where most blocks require a large number of samples. Note that several zones are evaluated as requiring 10,000 samples: for these zones, convergence is not reached in the reference image which was calculated with 10,000 samples and, in the context of this paper, we have stopped the calculations with this maximum value. The response of the model, having reached this limit, is that the block is still noisy. The image of the White-room night scene is another example of indirect lighting (the light source is included in the semi-transparent luminaire on the ceiling), for which the model proposes thresholds with important values, but lower than the maximum allowed. The underside of the Staircase scene is lit indirectly, but the wall there is covered with a wallpaper texture. The phenomenon of texture masking thus reduces the perception of noise and limits the number of samples needed. Also note the blocks 9 and 10 for the Staircase 2 scene, which require a relatively large number of samples. The convergence of the path tracing here is made more difficult due to the presence of a glass railing, which generates noise for a longer period of time. Overall, the results of the predictions obtained are fairly representative of the SSIM scores presented previously in Figure 16 and show that the model’s predictions appear correct on new image data.

## 7. Conclusions

The lighting simulation methods used in image synthesis provide images that are qualified as photorealistic and the power of these techniques makes them an increasingly popular tool for multimedia production companies. Nevertheless, these simulations are intrinsically subject to visual noise, which degrades the quality of the images obtained, due to their use of calculation methods based on Monte Carlo integration. The noise reduction requires an increase in the number of light path samples to be used, without a reliable criterion being available at the moment to define a threshold for stopping the calculations.

The work presented in this paper aims to propose an entirely generic method to establish a perceptual stopping criterion for the generation of this kind of image. They are based on the use of human perceptual thresholds obtained from a database of images representative of the images likely to be created today (The data presented in this study are available in https://prise3d.univ-littoral.fr). Our approach is based on a powerful linear algebra tool (SVD), the associated SVD entropy and their exploitation using RNN. This generic method gives good results for all types of images with diffuse and specular materials, as well as with direct and indirect illumination. It thus seems to offer a certain robustness, thanks to the coupling of the local information of the SVD-Entropy and the use of temporal processing of this information within a recurrent neural network. The use of SVD entropy as an input attribute seems to be more interesting than other previously published attributes, the use of which provides slightly inferior results.

Nevertheless, in some very rare cases, incorrect results are given by the proposed model. These cases require further investigation in order to understand their origin. We suppose that it lies in an under-representation of the light effects present in some blocks, which implies a defective learning in this case. Increasing the number of images for which the identified effects are under-represented should make it possible to eliminate these artifacts.

The good results obtained by the proposed approach encourage us to consider various research perspectives, aimed on the one hand at increasing the quality of the thresholds obtained, and on the other hand at ensuring their wider use. Improving the quality of the results can be investigated initially by identifying the contents of different image blocks (uniform or highly variable lighting, presence of numerous objects, etc.) and by generating a noise identification model specific to them. A simplification of the “models” to be learned would undoubtedly make it possible to obtain more accurate threshold predictions. Another related approach is to split the blocks according to their content, for example by segmentation techniques. This would make it possible to refine the identification of noise, which is not uniformly distributed in a block. From the point of view of extending the capabilities of the approach, a first avenue will consist in studying its generalization to other types of Monte Carlo integrators used in lighting simulation. We have initially restricted ourselves to path tracing, which is the most widely used method to date. It will be interesting to determine whether this method provides as good results for other integrators (i.e., bidirectional path tracing, metropolis light transport), or even whether it is able to manage all integrators with the same model. Finally, the extension of the approach to noise detection in stereoscopic images should also be considered. The singular value decomposition is particularly well suited to the management of rectangular matrices, that could be built by joining the images/blocks from both eyes. However, in this case, the problem of capturing human perceptual thresholds arises, and will require specific experimentations.

## Figures and Tables

**Figure 1 entropy-23-00075-f001:**
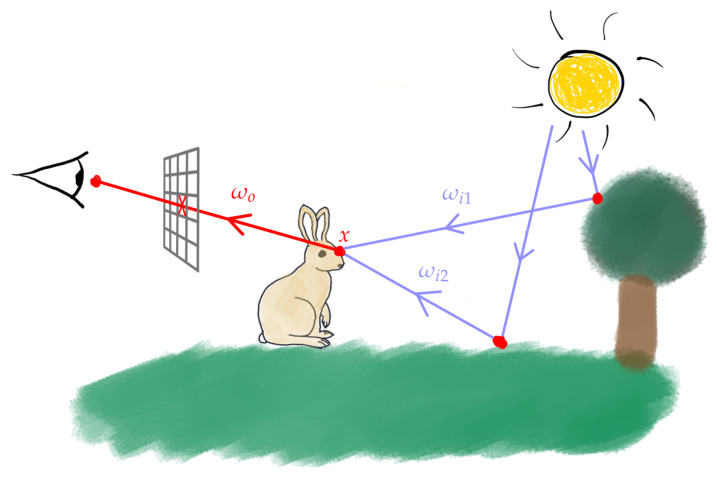
Some light paths thrown into a 3D scene from the light sources. In practice those paths are sampled in a reverse order, from the camera to the light sources.

**Figure 2 entropy-23-00075-f002:**
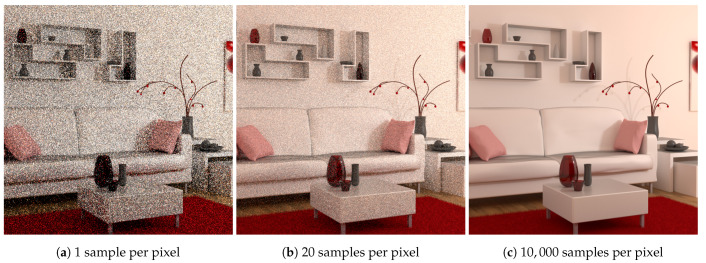
Some views of the image Living room 3 produced during paths sampling according to an increasing number of samples. Noise is highly visible on the two low sampled views.

**Figure 3 entropy-23-00075-f003:**
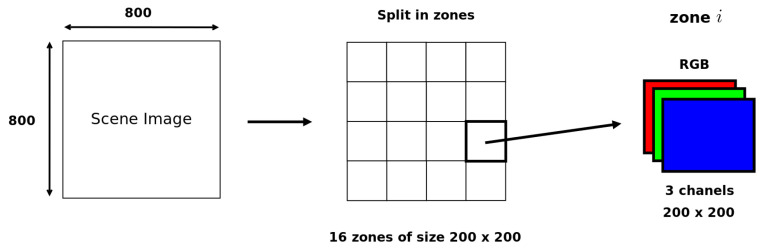
Division on the computed images of size 800×800.

**Figure 4 entropy-23-00075-f004:**
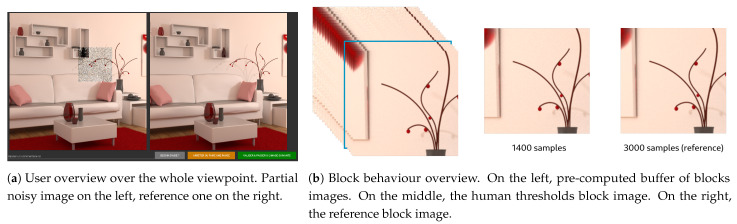
(**a**) offers an overview of user interface for the whole viewpoint. (**b**) describes how the user can change number of samples of the block: he can navigate through the pre-calculated images buffer to visually match the two images in this area. The blue bordered image in buffer, is the current selected image displayed in center.

**Figure 5 entropy-23-00075-f005:**
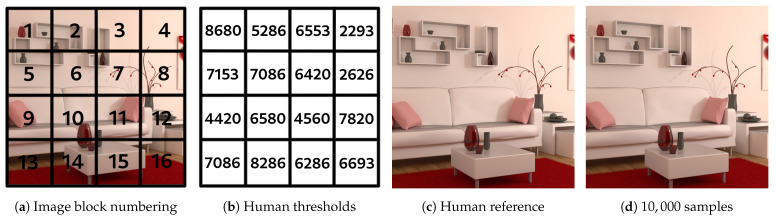
Reconstructed blocks of image and comparison of human image to reference image.

**Figure 6 entropy-23-00075-f006:**
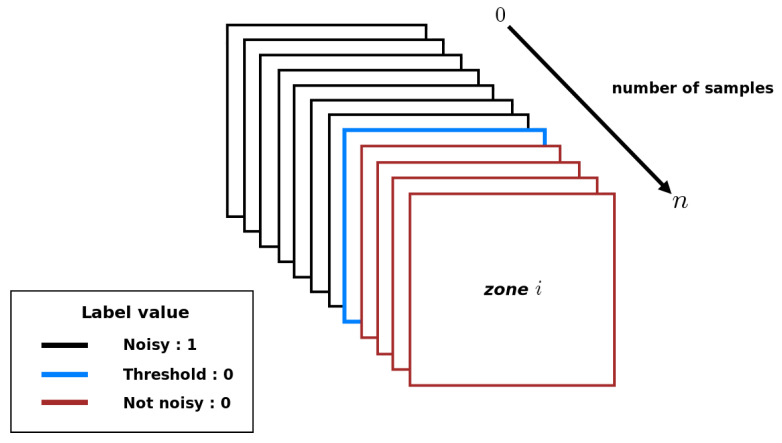
Label allocation for each sub-image according to the sampling level of a block of the image from 800×800.

**Figure 7 entropy-23-00075-f007:**
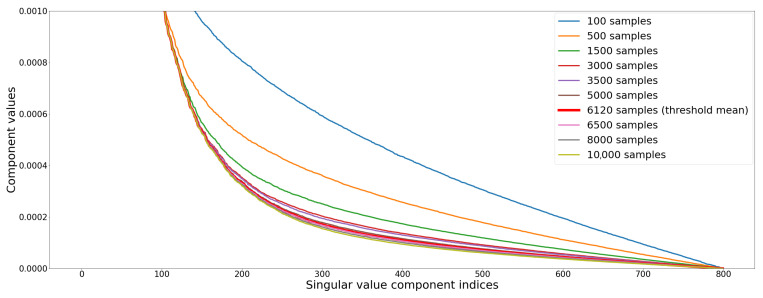
Σ components overview on Living room 3 scene, re-scaled to display low components.

**Figure 8 entropy-23-00075-f008:**
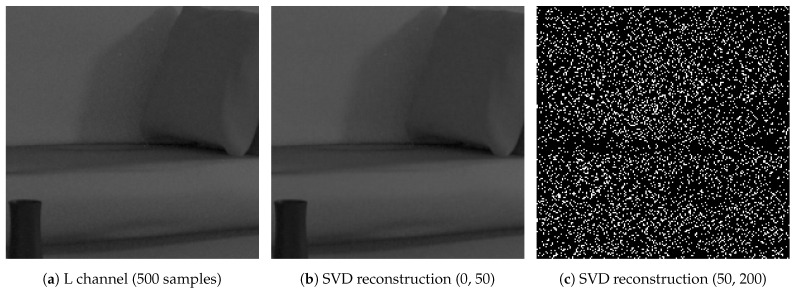
Singular Value Decomposition (SVD) reconstruction of a sub-image with size 200×200 of Living room 3 image by splitting the SV vector in two parts, the first quarter and remaining components of the SV vector.

**Figure 9 entropy-23-00075-f009:**
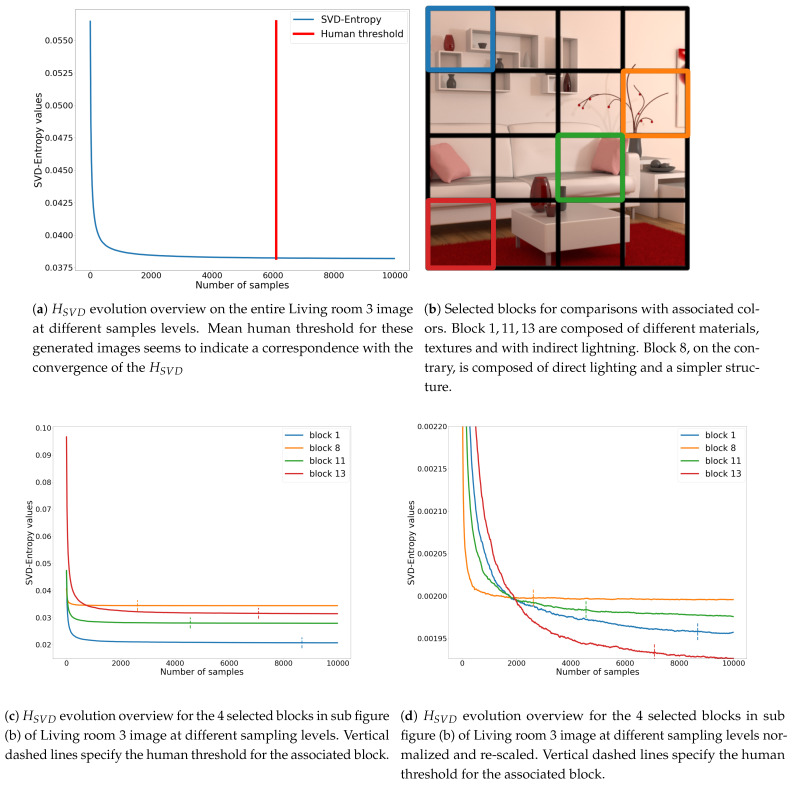
Overview of the evolution of $H_{SVD}$ for the Living room 3 image.

**Figure 10 entropy-23-00075-f010:**
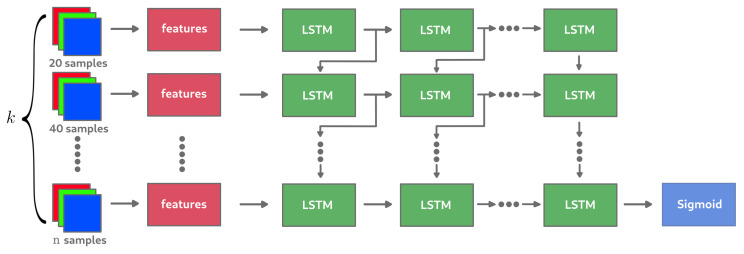
RNN expected architecture with *k* as sequence size. The additional sampling step between each image composing the sub-sequence is set here to 20. The images provided to the RNN in this example are therefore sampled with 20, 40, … (k−1)×20 samples.

**Figure 11 entropy-23-00075-f011:**
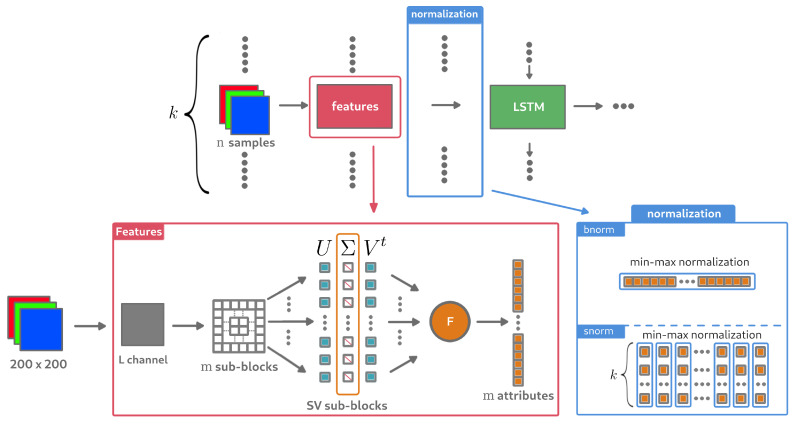
Fully pipeline with RNN architecture and associated tested parameters.

**Figure 12 entropy-23-00075-f012:**
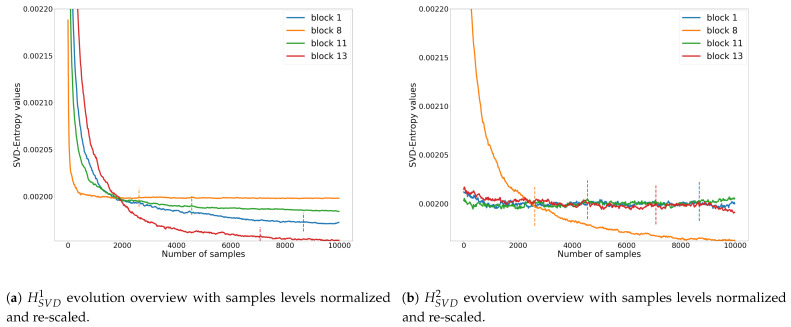
$H_{SVD}^{1}$ and $H_{SVD}^{2}$, high and low SVD components evolution overview for the 4 selected blocks in sub Figure 9b of Living room 3 image. Vertical dashed lines specify the human threshold for the associated block.

**Figure 13 entropy-23-00075-f013:**
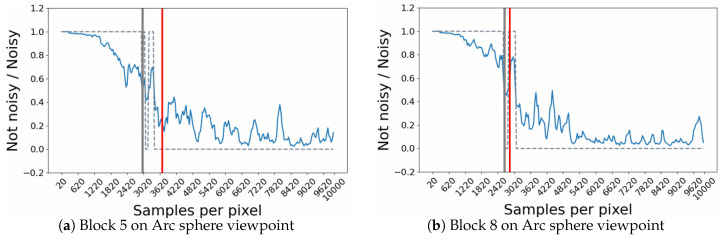
Some odel predictions during rendering of the selected RNN with $H_{SVD}$ inputs data. The red vertical line refers to the human threshold for the block. The blue curve corresponds to the model predictions during rendering. The blue dashed curve refers to the predicted label following the strict label selection criteria (If the probability is less than 0.5, then the image is considered to be no longer noisy, if not, it is still noisy). The vertical gray line, refers to the predicted thresholds obtained.

**Figure 14 entropy-23-00075-f014:**
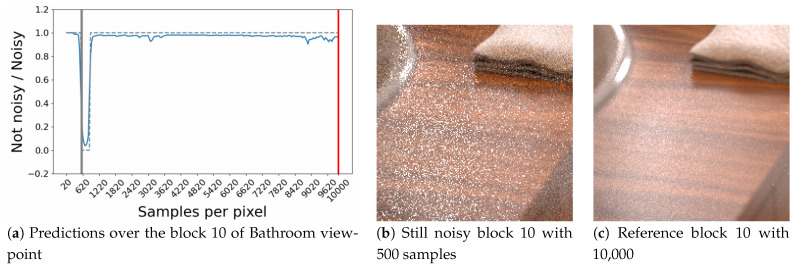
Case of critical threshold prediction obtained by the model. This case does not depend on the input data ($H_{SVD}$ or 26 attributes) but seems generalized to the RNN model. The blue curve corresponds to the model prediction during rendering. The blue dashed curve refers to the predicted label following the strict label selection criteria (If the probability is less than 0.5, then the image is considered to be no longer noisy, if not, it is still noisy). The vertical gray line, refers to the predicted thresholds obtained.

**Figure 15 entropy-23-00075-f015:**
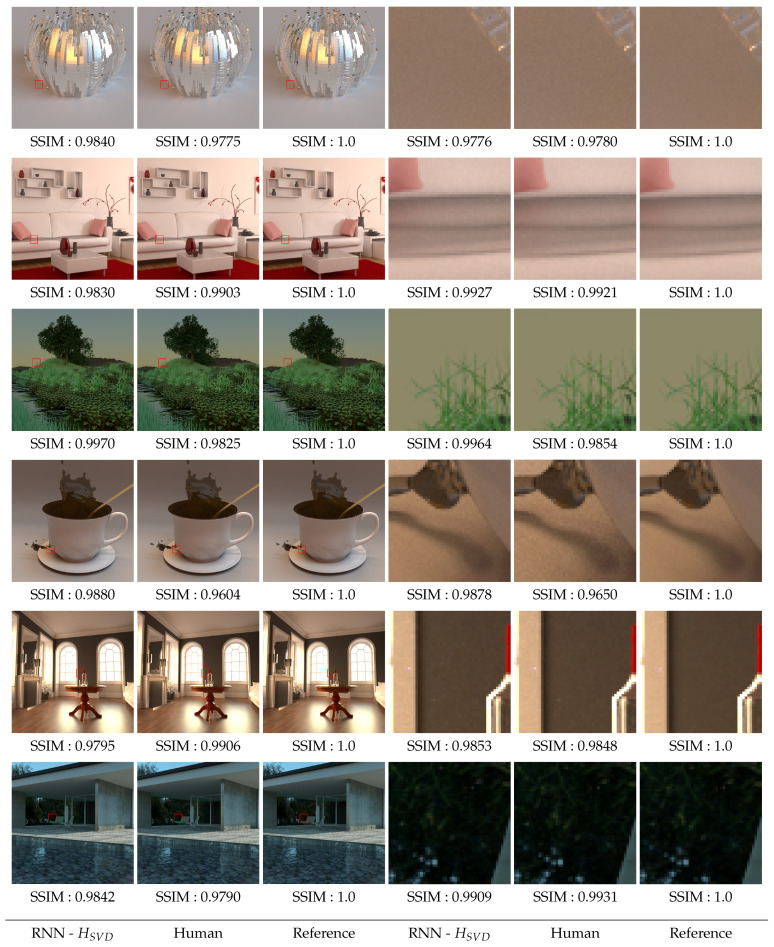
Reconstructed images from predicted thresholds using RNN with SVD-Entropy model and human thresholds. Each image is compared to the reference image using SSIM.

**Figure 16 entropy-23-00075-f016:**
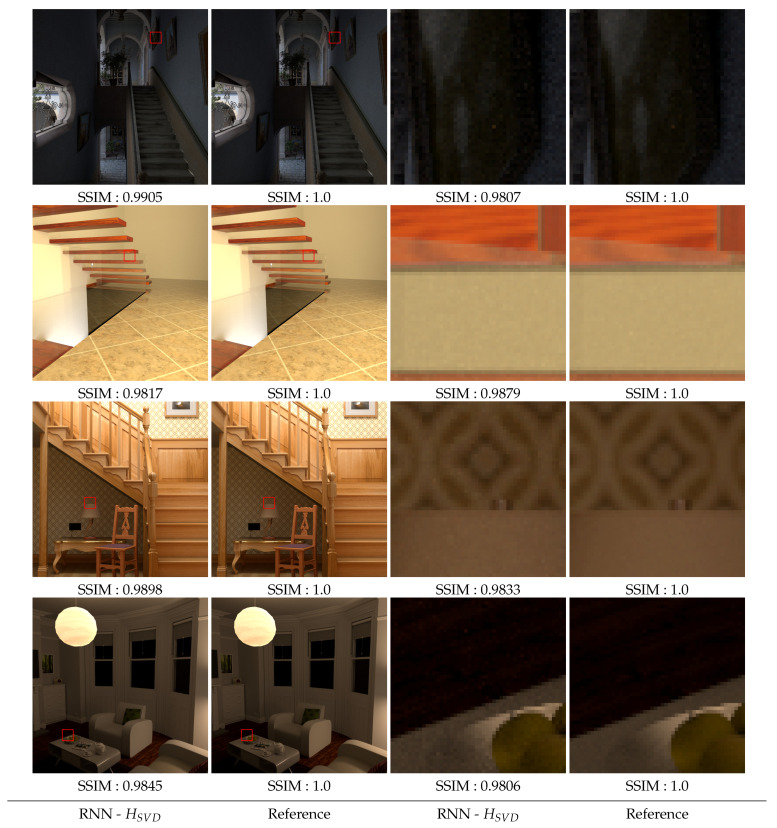
Reconstructed images from predicted thresholds using RNN with SVD-Entropy model over unlearned viewpoints compared to reference. Each image is compared to the reference image using SSIM.

**Figure 17 entropy-23-00075-f017:**
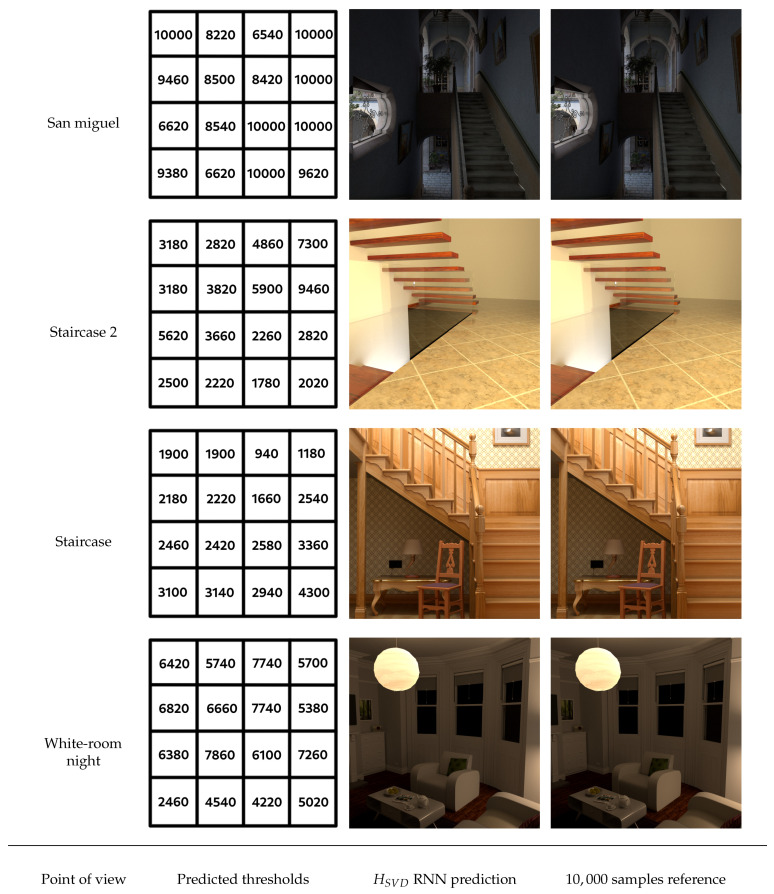
Predicted thresholds from $H_{SVD}$ RNN model and overview reconstructed blocks of 4 unlearned images with comparisons of each to the corresponding reference image.

**Table 1 entropy-23-00075-t001:** First 20 best models results for each combination of parameters, based on Area Under the Receiver Operating Curve (AUC ROC) score over test dataset. Acc means Accuracy metric over predictions.

*k*	*m*	*F*	$b_{s}$	bnorm	snorm	*n* Step	Acc Train	Acc Test	AUC Train	AUC Test
8	100	$H_{SVD}$	128	0	1	40	84.58%	82.74%	84.44%	82.55%
5	100	$H_{SVD}$	128	0	1	80	83.78%	82.87%	83.32%	82.47%
6	100	$H_{SVD}$	64	0	1	40	84.18%	82.61%	84.01%	82.45%
7	100	$H_{SVD}$	64	0	1	40	84.62%	82.79%	84.24%	82.44%
7	100	$H_{SVD}$	128	0	1	40	84.65%	82.75%	84.36%	82.42%
7	100	$H_{SVD}$	64	0	1	80	83.67%	82.36%	83.70%	82.38%
5	100	$H_{SVD}$	64	0	1	80	83.61%	82.17%	83.85%	82.27%
9	100	$H_{SVD}$	64	0	1	40	83.46%	81.99%	83.84%	82.21%
10	100	$H_{SVD}$	128	0	1	40	84.52%	82.58%	84.20%	82.16%
10	100	$H_{SVD}$	64	0	1	20	84.12%	82.16%	84.28%	82.15%
6	100	$H_{SVD}$	128	0	1	40	84.05%	82.25%	84.07%	82.13%
9	100	$H_{SVD}$	128	0	1	40	84.39%	82.82%	83.61%	82.10%
9	100	$H_{SVD}$	64	0	1	20	84.93%	82.48%	84.50%	82.06%
9	100	$H_{SVD}$	128	0	1	20	85.37%	82.56%	84.90%	82.04%
10	100	$H_{SVD}$	128	0	1	20	84.73%	82.53%	84.18%	82.02%
6	100	$H_{SVD}$	64	0	1	80	83.04%	81.68%	83.59%	82.01%
4	100	$H_{SVD}$	128	0	1	80	83.17%	81.81%	83.55%	82.00%
5	100	$H_{SVD}$	128	0	1	40	83.88%	82.23%	83.59%	81.95%
8	100	$H_{SVD}$	128	0	1	80	83.61%	81.89%	83.73%	81.88%
5	100	$H_{SVD}$	64	0	1	40	83.85%	82.02%	83.66%	81.84%

**Table 2 entropy-23-00075-t002:** First 20 best models Recurrent Neural Network (RNN) results with Constantin’s 26 attributes based on AUC ROC score over test dataset. Acc means Accuracy metric over predictions.

*k*	Batch	bnorm	snorm	*n* Step	Acc Train	Acc Test	AUC Train	AUC Test
9	128	0	1	20	81.16%	81.04%	81.29%	81.16%
8	64	0	1	20	81.14%	80.93%	80.85%	80.78%
10	128	0	1	20	80.42%	80.30%	80.92%	80.72%
9	64	0	1	20	80.17%	80.19%	80.66%	80.60%
7	128	1	1	20	82.14%	80.74%	81.92%	80.59%
10	128	1	1	20	81.67%	80.35%	81.96%	80.57%
10	64	0	1	20	80.57%	80.26%	80.68%	80.41%
10	128	1	1	40	81.78%	80.94%	80.56%	80.17%
9	128	1	1	20	82.24%	80.66%	81.59%	80.16%
9	64	1	1	20	80.95%	79.57%	81.43%	79.94%
7	128	0	1	20	79.44%	79.21%	80.33%	79.90%
8	128	1	1	40	80.57%	79.63%	80.67%	79.84%
8	64	1	1	40	82.07%	80.49%	81.14%	79.83%
10	128	0	1	40	79.43%	79.09%	80.09%	79.66%
10	64	1	1	40	80.45%	79.23%	80.76%	79.63%
6	128	1	1	20	81.96%	80.15%	80.96%	79.43%
8	64	1	1	20	81.19%	79.56%	81.06%	79.43%
9	128	1	1	40	80.72%	79.68%	80.11%	79.43%
6	64	1	1	40	80.08%	79.10%	80.35%	79.38%
8	128	1	1	20	80.64%	79.22%	80.81%	79.35%

**Table 3 entropy-23-00075-t003:** Threshold prediction error rate of models with an allowed margin of error on the 10,000 around the human threshold.

Model	$H_{\mathrm{SVD}}$	$H_{\mathrm{SVD}}$	$H_{\mathrm{SVD}}$	26 Attributes	26 Attributes	26 Attributes
**Margin** (in %)	2	6	10	2	6	10
**Global critical error (in %)**	14.79	13.25	11.89	17.82	16.21	14.73

## Data Availability

The data presented in this study are available in https://prise3d.univ-littoral.fr.

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
