# Peer review of "Stopping Criterion during Rendering of Computer-Generated Images Based on SVD-Entropy"

_entropy, 2021, doi:10.3390/e23010075_

Round 1

Reviewer 1 Report

The paper raises an essential issue of the estimation of image quality and noise perception in the field of photo-realistic computer graphics where noise is inherent in the calculation process. Unlike natural-scene images, however, a reference image is not available for computer-generated images.

The research design is appropriate. The results are presented. The results support the conclusions. The content of the paper is significant. The article is scientifically substantiated. It will find readers’ interest.

The introduction does not provide sufficient background and does not include all relevant references. Similar works were carried out by the Video Quality Experts Group (VQEG). The authors can find there both a project related to Computer Generated Imagery (CGI) and a project developing No-Reference models for visual quality (Monitoring of Audio-Visual Quality by Key Indicators, MOAVI). It would be worth referring to them.

The quality of the presentation is low. The methods are not adequately described. Charts/diagrams are (probably) raster graphics, which makes some fonts and details unreadable. It’s a good idea to put at least a paragraph of your introductory text into a section (for example, “2. Previous works”) before moving on to the first subsection (for example, “2.1. Improving the convergence of rendering”). Similarly, between section “6. Experimental results “and subsection” 6.1. The method parameters”.

Author Response

The authors would like to thank the Editors and the Reviewers for their constructive comments and helpful suggestions. In accordance to the comments made, the paper has been revised, updated, and improved. All the modifications requested by the reviewers are corrected in the new paper. Important changes and clarifications, in red color, are added to the paper. All changes and responses to comments are detailed in the PDF file provided in attachment.

Reviewer 2 Report

Please find the comment file in the attachement.

Author Response

(The authors gave the same response as above.)

Round 2

Reviewer 2 Report

Sufficient revision is done. The manuscript can be considered for publication.